# Contextual Image Masking Modeling via Synergized Contrasting without View Augmentation for Faster and Better Visual Pretraining

**Shaofeng Zhang**[1,2]**, Feng Zhu**[3]**, Rui Zhao**[1,3]**, Junchi Yan**[1,2]*
[1]MoE Key Lab of Artificial Intelligence, Shanghai Jiao Tong University
[2]Shanghai AI Laboratory     [3]SenseTime Group Limited
{sherrylone, yanjunchi}@sjtu.edu.cn
{zhufeng, zhaorui}@sensetime.com
Code: https://github.com/Sherrylone/ccMIM

## Abstract

We propose a new contextual masking image modeling (MIM) approach called contrasting-aided contextual MIM (ccMIM), under the MIM paradigm for visual pretraining. Specifically, we adopt importance sampling to select the masked patches with richer semantic information for reconstruction, instead of random sampling as done in previous MIM works. As such, the resulting patch reconstruction task from the remaining less semantic patches could be more difficult and helps to learn. To speed up the possibly slowed convergence due to our more difficult reconstruction task, we further propose a new contrastive loss that aligns the tokens of the vision transformer extracted from the selected masked patches and the remaining ones, respectively. The hope is that it serves as a regularizer for patch feature learning such that the image-level global information could be captured in both masked and unmasked patches, and notably such a single-view contrasting avoids the tedious image augmentation step required in recent efforts of introducing contrastive learning to MIM (to speedup convergence and discriminative ability). Meanwhile, the attention score from the contrastive global feature can also carry effective semantic clues to in turn guide our above masking patch selection scheme. In consequence, our contextual MIM and contrastive learning are synergetically performed in a loop (semantic patch selection-token alignment contrasting) to boost the best of the two worlds: fast convergence and strong performance on downstream tasks without ad-hoc augmentations, which are verified by empirical results on ImageNet-1K for both classification and dense vision tasks.

## 1 Introduction

Self-supervised learning (SSL) (Zbontar et al., 2021; Jin et al., 2022; Chen et al., 2020b) has been attracting increasing attention recently in deep learning, due to its label-free property and capability of learning rich holistic representations. Recent SSL methods mainly fall into two classes. **Contrastive** methods (He et al., 2020; Chen et al., 2020b; Zhang et al., 2021) construct multiple views of the given image to increase the variance and align the representations of these views in latent space. The pair-wise learning paradigm endows strong linear evaluation accuracy of contrastive methods. However, these methods tend to extract global information but ignore local information, which limits their performance on downstream dense vision tasks, e.g. detection, segmentation, etc. Besides, the two-view learning paradigm is usually sensitive to the choice of augmentation function (Chen et al., 2020c), batch size (Chen et al., 2020b; Zhang et al., 2021) and output dimension (Zbontar et al., 2021; Zhang et al., 2022) etc. With the success of recent ViT-based vision backbones, which divide images into several patches, **Mask-Image-Modeling (MIM)** methods (He et al., 2021; Fang et al., 2022; Chen et al., 2022) randomly mask some patches and use the self-attention mechanism to

---

*Junchi Yan is the correspondence author. Rui Zhao is also with Qing Yuan Research Institute, Shanghai Jiao Tong University. This work was in part supported by NSFC (62222607), Shanghai Municipal Science and Technology Major Project (2021SHZDZX0102) and SenseTime Collaborative Research Grant.

recover pixel-wise information from the remaining un-masked patches. These MIM-based methods are shown can obtain a better pre-trained encoder even than contrastive approaches, and especially show strong performance on transfer learning and finetuning tasks thanks to the ability to capture local information. However, these methods can suffer from the limited discriminability for global representation with less competitive linear accuracy (Gao et al., 2022), slow convergence speed (1,600 epochs for MAE), and GPU-consuming (e.g. with batch size 4096).

Therefore, recent attempts (Wang et al., 2022; Chen et al., 2022; Yi et al., 2022) are devoted to combining contrastive learning and MIM. The hope is that both (global) discriminative information (by contrasting) and spatial-sensitive information (by MIM) can be captured. Although these approaches show improved convergence speed (in the sense of fewer epochs needed for convergence) compared with using MIM alone, they can hardly further boost the performance of MIM on downstream tasks given even more (e.g. 800, 1600) pretraining epochs. Meanwhile, such direct combing contrasting with MIM can also bring about side effects e.g. the increased sensitivity to data augmentation and more overhead at each training epoch. In this paper, we aim to improve both convergence speed and accuracy by devising a new contextual MIM scheme, which is synergistically aided by contrastive learning. The highlights of our proposed ccMIM are:

**1) Novel framework for synergizing MIM and contrastive learning in a close-loop:** We propose a novel contextual MIM framework by actively selecting the rich semantic patches as masking patches for reconstruction to improve the MIM learning difficulty, whereby the semantic clue is learned and measured by our devised vision transformer token alignment contrasting loss. As such, the synergizing of the two components can be fulfilled in the first place. In contrast, existing efforts on the combination of MIM and contrasting often perform these two components independently until their weighted loss is finally computed and show limited accuracy improvement (perhaps also partly due to their random patch selection strategy as observed in our ablation study in Sec. 4.3).

**2) Cost-effective technical design for contextual MIM:** Under the above framework, we propose to use importance sampling to contextually select the patches with richer semantic information for masking and reconstruction, to increase the learning difficulty and effectiveness. The selection is guided by the attentional semantic score derived from the $[CLS]$ token alignment as contrastive learning between the selected and un-selected patches. Compared to the widely adopted two-view augmentation for contrasting in recent MIM works, our contrasting is efficiently performed within a single view (for speedup MIM training). Moreover, our way of using contrasting also directly guides the MIM learning instead of being two independent components as done in the existing literature.

**3) Improvement over MAE:** Using ViT-B (Dosovitskiy et al., 2020) as a standard protocol in MIM, ccMIM achieves 83.6%, 84.2% top-1 accuracy with 300 and 800 epochs pre-training, outperforming MAE (He et al., 2021) 0.8% (82.8% for 300 epochs) and 0.6% (83.6% for 1600 epochs) accuracy on ImageNet-1K, respectively. Source code will be made publicly available.

## 2 RELATED WORK

In our work, the proposed ccMIM divides image patches into two sets by information density by learning two objective functions. One is aligning the global representations of two sets and the other is reconstructing raw images from the visible set, so we briefly review contrastive learning (align representations of multiple augmented views in latent space) and the MIM-based methods (reconstruction).

**Contrastive learning** aims to learn instance discriminative representations to distinguish an image from the others (Hjelm et al., 2018). This is achieved by pulling together the representations of different views of the same image and pushing away the other images. Thus, most contrastive methods adopt siamese network (He et al., 2020; Grill et al., 2020; Chen & He, 2021). To create different views for the same image, well-designed data augmentation functions have been deployed (e.g., those investigated in SimCLR (Chen et al., 2020b)). To increase the number of negative pairs, Moco (He et al., 2020; Chen et al., 2020c) constructs a large queue to store negative representation in memory. Besides, Moco and BYOL use momentum and stop-gradient mechanisms are adopted to prevent degenerate solutions (Tian et al., 2021; Wang & Isola, 2020). To simplify BYOL, SimSiam (Chen & He, 2021) discards the momentum updating and finds stop-gradient and predictor head are the keys to preventing collapse. MoCo-v3 (Chen et al., 2021) and DINO (Caron et al., 2021) are based on the siamese network and extend MoCo (He et al., 2020) and BYOL (Grill et al., 2020) with Vision

Transformer (ViT) (Dosovitskiy et al., 2020) as their model backbones. Although contrastive learning methods provide discriminative features, most of them focus on learning global representations while lacking the spatial-sensitive and local representation that can be mitigated by the MIM techniques.

**MIM-based** methods (He et al., 2021; Xie et al., 2021) learn vision representation by reconstructing the masked patches from the partial observations. Based on the reconstruction objective, these methods can be divided into: pixel-wise reconstruction (He et al., 2021; Xie et al., 2021) and auxiliary features/tokens prediction (Dong et al., 2021; Chen et al., 2022). SimMIM (Xie et al., 2021) and MAE (He et al., 2021) are the first two methods applying mask modeling in the visual domain. They propose to reconstruct the raw pixel values from either the full set of image patches (mask tokens and visible patches for SimMIM) or partially observed patches (visible patches for MAE). Compared with SimMIM, MAE is more efficient because of dropping out a large portion of input patches. Inspired by adversarial training (Goodfellow et al., 2014), CIM (Fang et al., 2022) adds perturbations to raw images to enhance robustness for reconstruction. While GreenMIM (Huang et al., 2022) applies divide-and-conquer with hierarchical ViT backbones (Liu et al., 2021). Departure from predicting on RGB space, iBOT (Zhou et al., 2022) and AttMask (Kakogeorgiou et al., 2022) proposed to randomly and selectively (by semantic importance) mask patches and perform contrasts in latent space, respectively.

**Combination of Contrasting and MIM** has been recently explored with the hope to achieve both fast convergence and good accuracy. For instance, CAE (Chen et al., 2022) randomly partitions the image into two sets of patches and utilizes the visible set to reconstruct the masked patches. Then, an alignment objective is added to two sets in latent space. However, the reconstruction loss and alignment loss (as contrasting) are independent of each other and linearly combined for training, which we argue it lacks a synergistic effect between the two modules. Similar issues also exist in other attempts (Yi et al., 2022; Wang et al., 2022) with a linear combination of the two losses, although faster convergence is observed yet with little improvement on the accuracy of downstream tasks. In this paper, we aim to propose a more synergetic approach for contrasting-aided MIM.

## 3 METHODOLOGY

### 3.1 PRELIMINARIES

We denote the set of $N$ number of images $\mathbf{X} \in \mathbb{R}^{N \times C \times H \times W}$ and an encoder $f$ for pretraining.

**1) Contrastive Learning:** The encoders are usually quantified as CNNs and ViTs to learn global features $\mathbf{Z} \in \mathbb{R}^{N \times D}$ from raw images. The raw images are firstly transformed by certain augmentation to generate two views. The key step is using encoders to learn the invariance of two views, i.e. maximizing the representation similarity of the two views (alignment term in (Chen et al., 2020b; Wang & Isola, 2020)). However, directly adding the alignment part will cause degenerate solutions (Tian et al., 2021; Wang & Isola, 2020), which means all the images are encoded to the same and collapsed representation. Two techniques have been developed to avoid degenerate solutions, i.e., adopting stop-gradient to build asymmetric framework (Grill et al., 2020; He et al., 2020; Chen et al., 2020c; Caron et al., 2021; Zhou et al., 2022) and using negative samples (Chen et al., 2020b; He et al., 2020; Chen et al., 2020c), which will be used in the two versions of our method, which are respectively based on: the contrastive forms of SimSiam (Chen & He, 2021) and SimCLR (Chen et al., 2020b).

**i) Adopting the asymmetric framework:** the objective of SimSiam can be formulated as:

$$\mathcal{L}_{simsiam} = 2 - \text{sim}(\text{StopGrad}(\mathbf{H}^A), p(\mathbf{H}^B)) - \text{sim}(\text{StopGrad}(\mathbf{H}^B), p(\mathbf{H}^A)) \tag{1}$$

where $StopGrad(\cdot)$ means stop-gradient operations and $sim(\cdot, \cdot)$ is the similarity metric function (cosine similarity in this paper). $p(\cdot)$ is the predictor head (two layers MLP), which is commonly used in previous asymmetric methods (Chen & He, 2021; Grill et al., 2020). $\mathbf{H}^A$ and $\mathbf{H}^B$ denotes the representations of view $A$ and view $B$ in contrastive space, which can be obtained by $\mathbf{H} = g(\mathbf{Z})$ and $g$ is the projector head composed of two-layer MLP (LeCun et al., 2015).

**ii) Adopting the negative samples:** accordingly SimCLR (Chen et al., 2020b) adopts the negative-based objective: (InfoNCE (Hjelm et al., 2018)):

$$\mathcal{L}_{simclr} = -\mathbb{E}_{A,B} \left[ \log \frac{e^{sim(\mathbf{h}_i^A, \mathbf{h}_i^B)/\tau}}{\sum_{j=1, j \neq i}^N e^{sim(\mathbf{h}_i^A, \mathbf{h}_j^A)/\tau} + \sum_{j=1}^N e^{sim(\mathbf{h}_i^A, \mathbf{h}_j^B)/\tau}} \right] \tag{2}$$

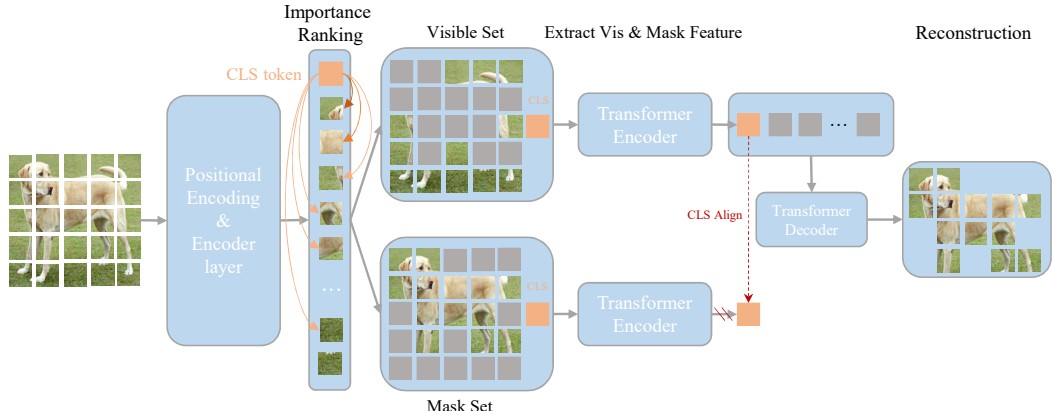

Figure 1: Framework of the proposed ccMIM. Stop gradient operation is added on the bottom transformer, since the mask set includes richer contextual information (similar to teachers).

where $\mathbf{h}_i^A$ ($\mathbf{h}_i^B$) means the representation in contrastive space of view $A$ ($B$) of image $i$.

**2) Masking Image Modeling:** the common embodiment of encoders are transformer-based models, e.g. vallina ViT (Dosovitskiy et al., 2020) and Swin Transformer (Liu et al., 2021). Current MIM-based methods (He et al., 2021; Chen et al., 2022; Xie et al., 2021) basically split raw images into several patches, and randomly select patches to mask. The remaining patches are used for reconstructing the masked patches $\hat{\mathbf{x}}_M$, and the pixel-wise objective is added on masked prediction:

$$\mathcal{L}_{recons} = \frac{1}{\Omega(\hat{\mathbf{x}}_M)} \|\hat{\mathbf{x}}_M - \mathcal{T}(\mathbf{x}_M)\|_p \tag{3}$$

where $\Omega(\hat{\mathbf{x}}_M)$ is the number of masked patches and $p$ means the $p$-norm ($p = 2$ in this paper). $\mathcal{T}$ is a transformation function (for MAE, $\mathcal{T}$ is the identity function; for MaskFeat (Wei et al., 2021), $\mathcal{T}$ is the HOG transformation). In this paper, we mainly follow the settings in MAE (He et al., 2021).

### 3.2 THE PROPOSED CCMIM: CONTRASTING-AIDED CONTEXTUAL MIM

In our proposed ccMIM, the patches are divided into two sets: masked set (composed of patches with richer contextual information) and visible set (with less contextual information, e.g. background). The visible set is used for reconstruction, and the contrastive objective is regularized on the global tokens of the two sets to help divide patches by contextual information. As illustrated in Fig. 1, ccMIM consists of five components:

• **Contextual Patch Modeling.** Given an input image, ccMIM first divides the patches into visible and masked sets by contextual information. Patches in the masked set will have richer contextual information to improve the difficulty of MIM. Both two sets will be fed to the backbone.

• **Encoder architecture.** Encoder backbone extracts a latent feature representation for the masked image, which is then used to predict the original signals at the masked area. The learned encoder is expected to be transferable to various vision tasks. In this paper, following MAE (He et al., 2021), we mainly consider ViT (Dosovitskiy et al., 2020) as backbones.

• **Projector head.** The projector head is applied in the contrastive branch, which forces the encoder to learn the global invariance of two sets (masked area and visible area). Specifically, we perform unidirectional alignment (Hjelm et al., 2018; Chen & He, 2021) on two sets (i.e., align visible set with masked set), since the masked set has richer contextual information.

• **Decoder.** The decoder will be applied to the latent feature representation of the visible set (with less contextual information) to produce one form of the original signals of the masked areas.

• **Prediction target.** This component defines the form of original signals to predict. It can be either the raw pixel values or a transformation of the raw pixels (HOG features).

In the following, we elaborate on the details of our approach.

**Selecting and dividing strategy.** We denote the image as $\mathbf{X} \in \mathbb{R}^{C \times H \times W}$. Under the ViT-based (Dosovitskiy et al., 2020) MIM paradigm, we first split each image to several patches $\mathbf{X} \in \mathbb{R}^{p^2 \times C \times \frac{H}{p} \times \frac{W}{p}}$. Then, each patch will be mapped to one vector through a linear layer $\mathbf{W} \in \mathbb{R}^{C\frac{H}{p}\frac{W}{p} \times D}$, where $D$ is the hidden dimension. Inspired by the transformer (Vaswani et al., 2017) in NLP, ViT adds one global token to extract global information of each image. Denote the $i$-th patch as $\mathbf{x}_i$ and let $\mathbf{x}_0$ be global token. Then, we can calculate the importance of the patches $\mathbf{x}_i$ by:

$$s = Softmax\left(\mathbb{E}_h\left[\frac{\mathbf{q}_0\mathbf{K}_{1:}^\top}{\sqrt{D}}\right]\right) \tag{4}$$

where $\mathbb{E}_h$ means taking the average attention value of multiple heads (Vaswani et al., 2017). $\mathbf{q}_0$ means the query vector of the global token, which can be calculated by $\mathbf{q}_0 = \mathbf{x}_o\mathbf{W}_Q$ and $\mathbf{W}_Q$ refers to the learnable weights. $\mathbf{K}_{1:}$ is the key matrix (Vaswani et al., 2017) excluding the first row (global token), which is calculated by $\mathbf{K} = \mathbf{X}\mathbf{W}_K$ and $\mathbf{W}_K$ is the learnable weights. After obtaining the importance of the patches, we can selectively divide the patches into two sets, i.e., masked set $\mathcal{M}$ and visible set $\mathcal{V}$, where $\mathcal{M}$ mainly includes the patches with larger importance and $\mathcal{V}$ is composed of the remaining patches. The main rationale behind this design is that some background patches (less attention value) may still have a little contextual information, which may be helpful for learning contextual and fine-grained information for downstream tasks (e.g., fine-grained classification, detection). Denote the probability density function of the importance is $f(x)$ and let $p(x)$ be a random distribution. Note $\pi(x)$ is the distribution of function $f$ on $p$. Then, for the expectation of $f(x)$ on $\pi$, we have:

$$\mathbb{E}[f] = \int_x \pi(x)f(x)dx = \int_x p(x)\frac{\pi(x)}{p(x)}f(x)dx \tag{5}$$

where $\frac{\pi(x)}{p(x)}$ is the importance sampling weights. For the new distribution $\frac{\pi(x)}{p(x)}f(x)$, we have:

$$Var_{x\sim p}\left[f(x)\frac{\pi(x)}{p(x)}\right] = \mathbb{E}_{x\sim p}\left[\left(f(x)\frac{\pi(x)}{p(x)}\right)^2\right] - \left(\mathbb{E}_{x\sim p}\left[f(x)\frac{\pi(x)}{p(x)}\right]\right)^2$$
$$= \mathbb{E}_{x\sim\pi}\left[f(x)^2\frac{\pi(x)}{p(x)}\right] - (\mathbb{E}_{x\sim\pi}[f(x)])^2 \tag{6}$$

where $\frac{\pi(x)}{p(x)}$ can be obtained by the observed importance values (calculated by self-attention) and random generator. Then, we can divide the patches into $\mathcal{M}$ and $\mathcal{V}$ sets by importance sampling.

**Visible and masked patches alignment.** For the importance masking strategy, one of the assumptions is the global token $[CLS]$ can learn the global information of the whole image. However, for the previous methods like MAE (He et al., 2021), there's no regularization added on the $[CLS]$ token. Hence, directly using $[CLS]$ to partition patches may still be random. To address this problem, we add another objective on the $[CLS]$ token, i.e. contrastive loss (InfoNCE (Chen et al., 2021) or $l_2$ regularization (Chen & He, 2021)). Denote the global representations ($[CLS]$ token) in contrastive space of masked set and visible set as $\mathbf{H}^\mathcal{M}$ and $\mathbf{H}^\mathcal{V}$, respectively. Then, the NCE-liked objective is:

$$\mathcal{L}_{NCE} = -\mathbb{E}_\mathcal{V}\left[\log\frac{e^{\text{sim}(\text{SG}(\mathbf{h}_i^\mathcal{M}),\mathbf{h}_i^\mathcal{V})/\tau}}{\sum_{j=1,j\neq i}^N e^{\text{sim}(\mathbf{h}_i^\mathcal{V},\mathbf{h}_j^\mathcal{V})/\tau} + \sum_{j=1}^N e^{\text{sim}(\text{SG}(\mathbf{h}_i^\mathcal{M}),\mathbf{h}_j^\mathcal{V})/\tau}}\right] \tag{7}$$

where $SG(\cdot)$ is the stop gradient operation. Compared with SimCLR-based objective, Eq. 7 only contrasts the visible branch. This design is because the masked set mainly includes the important patches, which have richer semantic information than the visible set. Hence, $\mathbf{h}^\mathcal{M}$ can be regarded as the teacher to guide the $\mathbf{h}^\mathcal{V}$ (student). We can also only add the alignment term on the $[CLS]$ token like SimSiam (Chen & He, 2021) and BYOL (Grill et al., 2020), which can be written as:

$$\mathcal{L}_{align} = \mathbb{E}\left[1 - \text{sim}(\text{StopGrad}(\mathbf{h}_i^\mathcal{M}), p(\mathbf{h}_i^\mathcal{V}))\right] \tag{8}$$

where $sim(\mathbf{x},\mathbf{y}) = \frac{\mathbf{x}^\top\mathbf{y}}{\|\mathbf{x}\|_2\|\mathbf{y}\|_2}$ in this paper and $p(\cdot)$ is the projector head (Chen & He, 2021).

**Image reconstruction and contrastive learning.** We adopt a standard The masked patch reconstruction protocol in ccMIM is similar to MAE, by predicting the pixel values for patches in the masked set. Each element in the decoder's output is a vector of pixel values representing a patch. The

last layer of the decoder is a prediction target $\mathbf{W} \in \mathbb{R}^{D \times \frac{H}{P} \cdot \frac{W}{P}}$. Then, in line with MAE, the mean squared error (MSE) is computed between the reconstructed and original patches in the pixel space:

$$\mathcal{L}_{recons} = \frac{1}{\Omega(\mathcal{M})} \|\hat{\mathbf{x}}^{\mathcal{M}} - \mathbf{x}^{\mathcal{M}}\|_2 \tag{9}$$

where $\Omega(\mathcal{M})$ is the number of elements in the masked set $\mathcal{M}$. The overall objective of ccMIM is:

$$\mathcal{L}_{ccMIM} = \begin{cases} \mathcal{L}_{recons} + \lambda \cdot \mathcal{L}_{align}, & \textit{for using asymmetric framework} \\ \mathcal{L}_{recons} + \lambda \cdot \mathcal{L}_{NCE}, & \textit{for using negative samples} \end{cases} \tag{10}$$

where $\lambda$ is the hyper-parameter to balance contrastive loss and reconstruction loss. Note that only using either of the contrastive loss (Eq. 7 and Eq. 8) is enough for regularizing the $[CLS]$ token. We mainly use $\mathcal{L}_{NCE}$ by default in our main experiments which we find can be even more stable. The comparison of the two objectives is given in the ablation study.

**Remark.** Note that though we also adopt the linear loss as in Eq. 10, while the reconstruction proce-dure is aided by the contrasting in a loop, which makes our method differs from other combination methods (Chen et al., 2022; Yi et al., 2022; Wang et al., 2022) without such synergetic connections.

## 3.3 METHODOLOGY COMPARISON

Note in Table 6 in Appendix, we compare recent SSL pretraining methods. In particular, here we elaborate on the connections to the most related methods CAE, MST, Repre, and SemMAE.

**Relation to CAE (Chen et al., 2022).** Both CAE and ccMIM partition the raw image into two sets, i.e., visible set and mask set. However, CAE divides them randomly, which is similar to MAE (He et al., 2021), limiting their performance. ccMIM partitions the two sets by importance sampling. Moreover, ccMIM adds another contrastive loss to regularize $[CLS]$ token to help partition.

**Relation to MST (Li et al., 2021).** Both MST and ccMIM use contrastive loss and reconstruction pretext task and selective strategy. However, the motivations of MST and ccMIM are completely different. MST mainly relies on the contrastive loss to learn the invariance of two views and reconstruction loss only works a little bit. Besides, they mask the unimportant patches, which only slightly influences their performance (Note that the random mask strategy will cause about 10% Top-1 accuracy drop). In contrast, ccMIM masks the important patches to enhance the difficulty of reconstruction. Besides, ccMIM does not require multiple views from tricky augmentation.

**Relation to Repre (Wang et al., 2022).** Repre is the other method using both contrastive loss and reconstruction loss. However, Repre also requires two views from tricky augmentation and the main gain is coming from the contrastive objectives. Besides, the reconstruction branch is similar to VAE (Kingma & Welling, 2013), which takes all patches as input without any masking strategy.

**Relation to SemMAE (Li et al., 2022).** SemMAE is another concurrent work about attentive masking. Both ccMIM and SemMAE mask patches with richer contextual information to increase the difficulty of reconstruction. SemMAE uses backbone pretrained by iBOT (Zhou et al., 2022) to help to select contextual regions, where iBOT takes 1600 extra epochs for pretraining, which is computation-cost. In contrast, ccMIM directly adds alignment loss on $[CLS]$ token to learn global information to select masked regions, which does not require extra information and time.

## 4 EXPERIMENTS

### 4.1 EXPERIMENTAL SETUP

**Dataset.** We perform self-supervised pre-training on the ImageNet-1K (Deng et al., 2009) training set, as commonly used in SSL (He et al., 2021; Zhang et al., 2021). It includes 1k classes which are well-balanced in distribution and the images contain an iconic view of objects. We evaluate our pretrained model on the ImageNet-1k validation set, as well as on MS-COCO (Lin et al., 2014) object detection and segmentation, ADE20K (Zhou et al., 2017) segmentation, and other small-scale classification datasets (Van Horn et al., 2018).

**Backbone.** We use the standard ViTs architecture (Dosovitskiy et al., 2020). It has several blocks composed of multi-head self-attention (MHSA) (Vaswani et al., 2017), MLP, and LayerNorm (Ba et al., 2016). ccMIM adds one $[CLS]$ token at both masked set and visible set (shared). The decoder

Table 1: Linear and finetune top-1 accuracy on ImageNet-1K. † means using the powerful multi-modal DALL·E (Reddy et al., 2021) tokenizer in pre-training. ‡ means methods from arXiv preprint to date.

| | Method | Model | #Params | PT Eps. | Linear | FT Eps. | FT Acc. (%) |
|---|---|---|---|---|---|---|---|
| Training from scrach | Scratch, DeiT (Touvron et al., 2021a) | ViT-B | 86M | 0 | - | 300 | 81.8 |
| | Scratch, MAE | ViT-B | 86M | 0 | - | 300 | 82.3 |
| | Scratch, Swin | Swin-B | 88M | 0 | - | 300 | 83.5 |
| Supervised Pre-training | Supervised, SimMIM | Swin-B | 88M | 300 | - | 100 | 83.3 |
| | Supervised, SimMIM | Swin-L | 197M | 300 | - | 100 | 83.5 |
| Contrastive Leanring | Moco V3 (Chen et al., 2021) | ViT-B | 86M | 800 | 76.5 | 100 | 83.2 |
| | DINO (Caron et al., 2021) | ViT-B | 86M | 400 | 77.3 | 100 | 83.3 |
| | iBOT (Zhou et al., 2022) | ViT-B | 86M | 1600 | 79.5 | 100 | 83.8 |
| Masked Image Modeling | MAE (He et al., 2021) | ViT-B | 86M | 300 | 62.4 | 100 | 82.8 |
| | MAE (He et al., 2021) | ViT-B | 86M | 1600 | 67.8 | 100 | 83.6 |
| | MaskFeat (Wei et al., 2021) | ViT-B | 86M | 300 | - | 100 | 83.6 |
| | BEiT† (Bao et al., 2021) | ViT-B | 86M | 300 | 37.6 | 100 | 83.0 |
| | CAE†,‡ (Chen et al., 2022) | ViT-B | 86M | 300 | 64.2 | 100 | 83.3 |
| | CAE†,‡ (Chen et al., 2022) | ViT-B | 86M | 800 | 68.3 | 100 | 83.6 |
| | CIM†,‡ (Fang et al., 2022) | ViT-B | 86M | 300 | - | 100 | 83.3 |
| | CIM‡ (Fang et al., 2022) | ViT-B | 86M | 300 | - | 100 | 83.1 |
| | ConMIM‡ (Yi et al., 2022) | ViT-B | 86M | 300 | - | 100 | 83.5 |
| | ConMIM‡ (Yi et al., 2022) | ViT-B | 86M | 800 | - | 100 | 83.7 |
| | ccMIM (Ours) | ViT-B | 86M | 300 | 66.9 | 100 | 83.6 |
| | ccMIM (Ours) | ViT-B | 86M | 800 | **68.9** | 100 | **84.2** |

of ccMIM follows the settings of MAE (He et al., 2021), which includes two-layer MHSA blocks. At the evaluation stage, the average embedding of patches is used for finetuning and linear probing.

**Evaluation protocols. i)** Linear probing is widely used as a proxy of pretraining quality evaluation for SSL (Chen et al., 2020b; He et al., 2021; Chen et al., 2022). It learns a linear classifier over the image-level representation output from the pretrained encoder by using the labels of the images, and then tests the performance on the validation set. **ii)** Fine tuning is often used to evaluate the backbone in reconstructed-based methods (He et al., 2021; Chen et al., 2022). We use the same hyper-parameter with MAE, i.e., 100 epochs with learning rate 1e-3 and 1024 batch size. **iii)** Downstream tasks across different datasets and tasks (detection, segmentation, fine-grained classification) are also used to evaluate the transferability of the pre-trained encoder.

## 4.2 MAIN RESULTS

**Linear probing and finetune on ImageNet-1K.** Table 1 shows the linear probing and fine-tuning accuracy on ImageNet-1k dataset. We mainly compare our method with transformer-based contrastive learning methods (Caron et al., 2021; Chen et al., 2021) and MIM-based (He et al., 2021; Xie et al., 2021) methods. For 300 epochs pretraining, our ccMIM outperforms baseline MAE **+0.8%** top-1 accuracy. For 800 epochs pretraining, ccMIM achieves 84.2% top-1 accuracy, outperforms MAE **+0.6%** accuracy with 1600 epochs pretraining. For the concurrent method ConMIM, it uses two views by non-trivial augmentation in the pre-training step, i.e., multiple views are required to enable contrastive objectives. Although ccMIM also has the contrastive objective, it only requires a single view. Besides, ConMIM gets 83.7% accuracy with 800 epochs pretraining, only improves 0.2% accuracy on the basis of 300 epochs pretraining. The reason may be that the contrastive objective in ConMIM only improves the convergence speed but not the final performance. In contrast, ccMIM gets 84.2% (0.6% higher than MAE and ccMIM with 300 epochs pretraining) top-1 accuracy in 800 epochs pretraining, which may be because the contrastive objective in ccMIM is to help select contextual patches and the final gain is coming from recovering patches with richer contextual information (ConMIM randomly masks patches with less contextual information). Besides, we also find ccMIM surpasses MAE **+4.5%** and **+2.1%** top-1 accuracy under linear probing protocol. We guess that's because ccMIM adds the global-discriminative objective (see ablation study in Sec. 4.3).

**Object detection and segmentation on MS-COCO. Setups.** Following CAE (Chen et al., 2022), we adopt Mask R-CNN (He et al., 2017) that produces bounding boxes and instance masks simultaneously, with the ViT as the backbone (See the supplementary file for training details). We apply the same object detection system to the methods. We report the box AP for object detection and the mask AP for instance segmentation. We utilize multi-scale training and resize the image with the size of the short side between 480 and 800 and the longer side no larger than 1,333. The batch size is 32. For ViT-S, the learning rate is 3e-4, the layer-wise decay rate is 0.75, and the drop path rate is 0.1. For ViT-B, the learning rate is 3e-4, the layer-wise decay rate is 0.75, and the drop path

Table 2: Object detection and instance segmentation on MS-COCO. Mask R-CNN is adopted and trained with the 1x schedule. All the results are based on the same implementation for object detection and instance segmentation. 'Epoch' refers to the number of pretraining epochs on ImageNet-1K. 'SSL?' means whether use self-supervised pre-training.

| Method | Backbone | Epoch | #Param. | SSL? | Detection $AP^{bb}$ | $AP^{bb}_{50}$ | $AP^{bb}_{75}$ | Segmentation $AP^{mk}$ | $AP^{mk}_{50}$ | $AP^{mk}_{75}$ |
|---|---|---|---|---|---|---|---|---|---|---|
| DeiT (Touvron et al., 2021a) | ViT-S/16 | 300 | 22M | × | 43.1 | 65.2 | 46.6 | 38.4 | 61.8 | 40.6 |
| MoCo V3 (Chen et al., 2021) | ViT-S/16 | 300 | 22M | ✓ | 43.3 | 64.9 | 46.8 | 38.8 | 61.6 | 41.1 |
| BEiT (Bao et al., 2021) | ViT-S/16 | 300 | 22M | ✓ | 35.6 | 56.7 | 38.3 | 32.6 | 53.3 | 34.2 |
| CAE (Chen et al., 2022) | ViT-S/16 | 300 | 22M | ✓ | 44.1 | 64.6 | 48.2 | 39.2 | **61.4** | 42.2 |
| ccMIM (Ours) | ViT-S/16 | 300 | 22M | ✓ | **44.3** | 64.9 | **48.3** | **39.5** | 61.4 | **42.4** |
| DeiT (Touvron et al., 2021a) | ViT-B/16 | 300 | 86M | × | 46.9 | 68.9 | 51.0 | 41.5 | 65.5 | 44.4 |
| Moco V3 (Chen et al., 2021) | ViT-B/16 | 300 | 86M | ✓ | 45.5 | 67.1 | 49.4 | 40.5 | 63.7 | 43.4 |
| DINO (Caron et al., 2021) | ViT-B/16 | 300 | 86M | ✓ | 46.8 | 68.6 | 50.9 | 41.5 | 65.3 | 44.5 |
| BEiT (Bao et al., 2021) | ViT-B/16 | 300 | 86M | ✓ | 39.5 | 60.6 | 43.0 | 35.9 | 57.7 | 38.5 |
| BEiT (Bao et al., 2021) | ViT-B/16 | 800 | 86M | ✓ | 42.1 | 63.3 | 46.0 | 37.8 | 60.1 | 40.6 |
| MAE (He et al., 2021) | ViT-B/16 | 300 | 86M | ✓ | 45.4 | 66.4 | 49.6 | 40.6 | 63.4 | 43.7 |
| MAE (He et al., 2021) | ViT-B/16 | 1600 | 86M | ✓ | 48.4 | 69.4 | 53.1 | 42.6 | 66.1 | 45.9 |
| ConMIM (Yi et al., 2022) | ViT-B/16 | 800 | 86M | ✓ | 48.7 | ∼ | ∼ | 43.6 | ∼ | ∼ |
| CAE (Chen et al., 2022) | ViT-B/16 | 300 | 86M | ✓ | 48.4 | 69.2 | 52.9 | 42.6 | 66.1 | 45.8 |
| CAE (Chen et al., 2022) | ViT-B/16 | 800 | 86M | ✓ | 49.8 | 70.7 | 54.6 | 43.9 | 67.8 | 47.4 |
| ccMIM (Ours) | ViT-B/16 | 300 | 86M | ✓ | 49.1 | 69.8 | 53.3 | 43.1 | 66.4 | 46.2 |
| ccMIM (Ours) | ViT-B/16 | 800 | 86M | ✓ | **50.3** | **71.2** | **55.0** | **44.5** | **68.4** | **47.9** |

Table 3: Semantic segmentation on ADE20K dataset. #Epochs means pretraining epochs on ImageNet-1K. #Views means used views for pretraining. #Iter is the iterations for finetuning.

| Method | Backbone | Arch | #Epochs | #Views | Supervised | Self-supervised | #Iter | mIoU | aAcc | mAcc |
|---|---|---|---|---|---|---|---|---|---|---|
| Moco V2 (Chen et al., 2020c) | ResNet50 | FPN | 200 | 2 | × | ✓ | 40k | 35.8 | 77.6 | 45.1 |
| DeiT (Touvron et al., 2021a) | ViT-B/16 | FPN | 300 | 1 | ✓ | × | 160k | 47.0 | ∼ | ∼ |
| Moco V3 (Chen et al., 2021) | ViT-B/16 | UperNet | 300 | 2 | × | ✓ | 160k | 46.2 | 56.4 | 82.4 |
| DINO (Caron et al., 2021) | ViT-B/16 | UperNet | 300 | 2 | × | ✓ | 160k | 45.0 | 55.1 | 82.1 |
| MAE (He et al., 2021) | ViT-B/16 | UperNet | 300 | 1 | × | ✓ | 160k | 45.2 | 55.2 | 81.9 |
| MAE (He et al., 2021) | ViT-B/16 | UperNet | 1600 | 1 | × | ✓ | 160k | 46.9 | 57.4 | 82.7 |
| CAE (Chen et al., 2022) | ViT-B/16 | UperNet | 300 | 1 | × | ✓ | 160k | 46.0 | 56.6 | 82.5 |
| CAE (Chen et al., 2022) | ViT-B/16 | UperNet | 800 | 1 | × | ✓ | 160k | 47.2 | 57.9 | 83.0 |
| ccMIM (Ours) | ViT-B/16 | UperNet | 300 | 1 | × | ✓ | 160k | 46.5 | 56.9 | 82.8 |
| ccMIM (Ours) | ViT-B/16 | UperNet | 800 | 1 | × | ✓ | 160k | **47.7** | **58.3** | **83.5** |

Table 4: Transfer learning on fine-grained datasets. All the methods are under the same settings.

| Dataset | ClusterFit | SNCA+ | Grafit | DINO | Sup. | iBOT | BeiT | MAE (1600 eps) | ccMIM (800 eps) |
|---|---|---|---|---|---|---|---|---|---|
| iNaturalist$_{2018}$ | 49.7 | 69.2 | 69.8 | 72.6 | 73.2 | 74.6 | 72.3 | 75.4 | **75.6** |
| iNaturalist$_{2019}$ | 73.7 | 74.5 | 75.9 | 78.6 | 77.7 | 79.6 | 79.2 | 80.5 | **81.0** |
| Flowers-102 | ∼ | 98.2 | 98.2 | 98.8 | 98.4 | 98.9 | 98.0 | 99.0 | **99.2** |

rate is 0.2. We train the network with the 1x schedule 12 epochs with the learning rate decayed by 10x at epochs 9 and 11. **Results.** Table 2 shows the results on COCO detection and segmentation. ccMIM outperforms previous MIM-based methods with a large range. Specifically, with 300 epochs pretraining, ccMIM surpasses MAE **+3.7%** $AP^{bb}$ and **+2.5%** and $AP^{mk}$ on detection and segmentation, respectively. With 800 epochs pretraining, ccMIM outperforms MAE **+2.5%** $AP^{bb}$ $AP^{bb}$ and **+1.9%** and $AP^{mk}$. We also find ccMIM surpasses CAE **+0.5%** $AP^{bb}$ and **+0.6%** $AP^{mk}$ with 800 epochs pretraining, while only surpasses **+0.2%** $AP^{bb}$ and **+0.3%** $AP^{mk}$ on ViT-S backbone, which we guess that's because the reconstruction in ccMIM (attentive selection) is more difficult than in CAE (random). Hence, ccMIM has better improvements when using larger backbones.

**Semantic segmentation on ADE20K. Setups.** We evaluate semantic segmentation performances of the pre-trained models on ADE20K (Zhou et al., 2017), which includes 150 fine-grained semantic categories and 25k training data. In line with iBOT (Zhou et al., 2022), we report the results by three metrics: mean intersection of union (mIoU) averaged over all semantic categories, all pixel accuracy (aAcc), and mean class accuracy (mAcc). We pretrain ccMIM with 800 epochs and finetune it in FPN (Lin et al., 2017) with batch size 16. For ViT-B, the layer-wise decay rate is 0.65, and the drop path rate is 0.1. For other methods, we download their pretrained models and finetune them in our setting for fair comparisons and Table 3 shows the finetune results. The proposed ccMIM gets state-of-the-art accuracy among all the contrastive-based and MIM-based methods. Specifically, for 800 epochs pretraining, ccMIM outperforms MAE and CAE **+1.7%** and **+0.5%** mIoU, respectively.

**Transfer classification on other datasets.** In line with previous SSL methods, we conduct a set of experiments to evaluate the transfer accuracy on fine-grained classification datasets (iNaturalist (Van Horn et al., 2018) and Flowers). We mainly compare our method with ClusterFit (Yan et al., 2020), SNCA+ (Wu et al., 2018), Grafit (Touvron et al., 2021b), iBOT (Zhou et al., 2022), DeiT (Touvron et al., 2021a) and MAE (He et al., 2021). We finetune the pretrained encoder on iNaturalist datasets in 360 epochs with the learning rate 3e-4. For all SSL methods, we choose ViT-B/16 for comparisons. The results in Table 4 show that ccMIM obtains the highest accuracy compared with previous supervised and self-supervised methods. Specifically, ccMIM gets 0.5%, 0.2% and 0.2% higher accuracy than MAE on iNaturalist$_{2019}$, iNaturalist$_{2018}$ and Flowers-102, respectively.

## 4.3 ABLATION STUDY

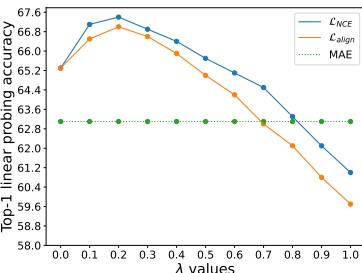

Figure 2: Impact of different $\lambda$.

**Regularization on classification token and effect of $\lambda$.** Different from ConMIM (Yi et al., 2022), ccMIM does not require multiple views for contrasting and the main improvement is due to the contrasting-based contextual MIM. To verify this, we try different $\lambda$ on both $\mathcal{L}_{align}$ and $\mathcal{L}_{NCE}$ in experiments. In detail, we try $\lambda$ from 0 (no regularization) to 1.0 by fixing training epochs as 400 and other hyper-parameters (e.g., learning rate as 1e-5, batch size as 1024). We report top-1 linear probing accuracy in Fig. 4.3. We find that $\mathcal{L}_{NCE}$ and $\mathcal{L}_{align}$ has similar properties (they gets best results when $\lambda = 0.2$). When $\lambda > 0.2$, with the increase of $\lambda$, the accuracy drops. When $\lambda > 0.8$, ccMIM with $\mathcal{L}_{NCE}$ gets lower accuracy than MAE, which we think this phenomenon is because the gain of ccMIM is from contextual masking, and contrastive loss is only used for selection semantic-richer patches, but not for direct improvement. When $\lambda = 0$, there's no regularization on $[CLS]$ token, leading the lower accuracy than $\lambda = 0.1$ and $\lambda = 0.2$.

Table 5: Ablation study of the dividing strategy.

| Dividing strategy | 100 epochs | | 400 epochs | |
|---|---|---|---|---|
| | Linear | Finetune | Linear | Finetune |
| random | 33.6 | 79.4 | 63.1 | 82.9 |
| block-wise | 32.9 | 79.2 | 62.0 | 82.8 |
| importance | 31.1 | 78.8 | 66.8 | 83.6 |
| importance sampling | **41.1** | **80.5** | **67.4** | **83.8** |

**Dividing strategy.** We further conduct a group of experiments to analyze the effect of different dividing strategies, i.e., random dividing, block-wise dividing, importance-based dividing, and importance-sampling-based dividing. For block-wise masking, we follow BEiT (Bao et al., 2021). Specifically, a block of image patches is masked each time. For each block, we set the minimum number of patches to 16. Then we randomly choose an aspect ratio for the masking block. We repeat the above two steps until obtaining enough masked patches. We distributed pretrain ccMIM on 32 GPUs in 100 and 400 epochs with 1024 batch size. We report linear and finetune accuracy in Table 5. In our experiments, the block-wise dividing strategy gets a bit lower accuracy than random dividing, which is in line with previous methods MAE (He et al., 2021) and SimMIM (Xie et al., 2021). We also find with 100 epochs pretraining, the importance-based dividing strategy gets much lower accuracy than random and block-wise methods. However, for 400 epochs pretraining, importance-based methods get higher accuracy than random and block-wise methods. We think that's because the importance-based strategy will slow down the convergence speed (patches with richer contextual information are more difficult to reconstruct). Importance-sampling-based method outperforms other dividing methods in both 100 and 400 epochs pretraining. We guess that's because, for the first few epochs, the variance of attention value of different patches is small. Thus, when adding the random sampling operation, ccMIM also divides some patches with less contextual information in the masked set. Then, the reconstruction task is not too difficult, leading to a faster convergence speed than the importance-based strategy. Then, for 400 epochs pretraining, $[CLS]$ token could easily divide patches with richer contextual information into the masked set, leading to the higher accuracy (**+4.3%**) than random and block-wise methods.

## 5 CONCLUSION

We have presented a novel contextual MIM framework dubbed ccMIM, whereby masked patch selection is aided by a contrast between the masked patches and unmasked ones. The reconstruction difficulty is increased as the masked patches are more semantically rich than random sampling as done in peer MIM methods. The resulting combined approach shows notable performance gain in both convergence speed and accuracy compared to baselines on public benchmarks across different tasks from classification to dense tasks including detection and segmentation.

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

Table 6: Methodology comparison for recent SSL pre-training methods. $CE$, $NCE$, $l_1$ and $l_2$ mean cross-entropy loss, infoNCE loss, $l_1$ distance loss and $l_2$ distance loss, respectively. 'Aug twice' means augmenting the images twice to learn the invariance of the two augmented views. "Extra knowledge" means whether use extra data or pretrained model.

| Method | Extra knowledge? | Aug twice? | Contrastive | Reconstruct | Mask strategy | Objectives |
|---|---|---|---|---|---|---|
| DINO (Caron et al., 2021) | ✗ | ✔ | ✔ | ✗ | – | $CE$ |
| iGPT (Chen et al., 2020a) | ✗ | ✗ | ✗ | ✔ | random | $CE$ |
| BeiT (Bao et al., 2021) | ✗ | ✗ | ✗ | ✔ | random | $CE$ |
| MST (Li et al., 2021) | ✗ | ✔ | ✔ | ✔ | selective | $CE$ & $l_1$ |
| iBOT (Zhou et al., 2022) | ✗ | ✔ | ✔ | ✗ | random | $CE$ |
| MAE (He et al., 2021) | ✗ | ✗ | ✗ | ✔ | random | $l_2$ |
| SimMIM (Xie et al., 2021) | ✗ | ✗ | ✗ | ✔ | random | $l_1$ |
| PeCo (Dong et al., 2021) | ✗ | ✗ | ✗ | ✔ | random | $CE$ |
| Repre (Wang et al., 2022) | ✗ | ✔ | ✔ | ✔ | – | $NCE$ & $l_1$ |
| MaskFeat (Wei et al., 2021) | ✗ | ✗ | ✗ | ✔ | random | $l_2$ |
| CIM (Fang et al., 2022) | ✗ | ✗ | ✗ | ✔ | random | $CE$ |
| CAE (Chen et al., 2022) | ✔ | ✗ | ✗ | ✔ | random | $l_2$ |
| ConMIM (Yi et al., 2022) | ✗ | ✔ | ✔ | ✔ | random | NCE |
| SemMAE (Li et al., 2022) | ✔ | ✗ | ✔ | ✔ | selective | $l_2$ |
| ccMIM (Ours) | ✗ | ✗ | ✔ | ✔ | selective | $l_2$ |

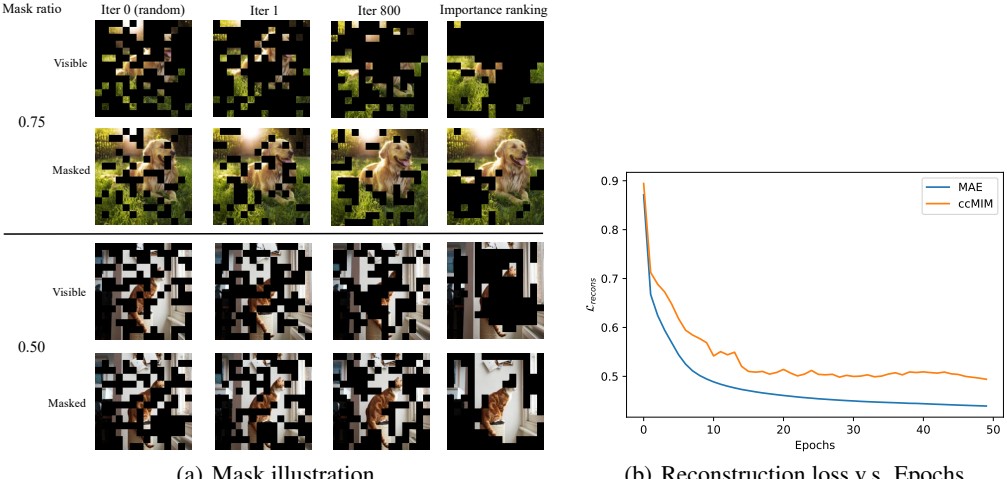

(a) Mask illustration      (b) Reconstruction loss v.s. Epochs

Figure 3: Left figure illustrates the patches division strategy with different masking ratios, where visible set mainly includes semantic-less patches to increase the difficulty of reconstruction. Right plot shows the $\mathcal{L}_{recons}$ in Eq. 9 v.s. epochs.

## A   METHODOLOGY COMPARISON

To better position our approach, we list recent contrastive and mask image modeling methods in Tab. 6.

## B   MASK ILLUSTRATION

For better understanding of our work, we illustrate how ccMIM divides the patches into visible and masked set in Fig. 3(a). We visualize four pretrained models, i.e., ccMIM with iteration 0, 1, 800 and importance-based dividing in Sec. 4.3. When $Iter = 0$, the $[CLS]$ token is randomly initialized, ccMIM randomly divides the patches into masked set and visible set. After 800 epochs pretraining, ccMIM could divide patches with richer contextual information into mask set, and the remain patches with less contextual information are divided into visible set to increase the difficulty of reconstruction. We further illustrate reconstruction loss curve of MAE and ccMIM in first 50 epochs pretraining in Fig. 3(b). We can find the loss values of ccMIM is higher than MAE, which demonstrates the larger difficulty for reconstruction of ccMIM than MAE.

## C  IMPLEMENTATION DETALS

**Pre-training hyper-parameters.** In line with MAE, we use AdamW as optimizer with the base learning rate 1.5e-4 and weight decay 0.05. We set the balance ratio $\lambda = 0.2$ and use $\mathcal{L}_{NCE}$ in Eq. 7 as regularization loss as default. For computational tractability, we set batch size as 1024, distributed pretraining on 32 NVIDIA 1080Ti GPUs. Similar to previous methods in SSL (Xie et al., 2021), we adopt cosine $lr$ scheduler and only use the random resized crop for augmentation.

