# OpenReview forum: "Contextual Image Masking Modeling via Synergized Contrasting without View Augmentation for Faster and Better Visual Pretraining"
_ICLR.cc/2023/Conference — ICLR 2023 poster_

### Official Review · Reviewer_tKmv · 2022-10-23

**Confidence:** 3
**Clarity, Quality, Novelty And Reproducibility:** Clarity, Quality, Novelty, and Reprod…
**Correctness:** 3
**Technical Novelty And Significance:** 3
**Empirical Novelty And Significance:** 3
**Recommendation:** 6

**Strength And Weaknesses:**

Strength
1, Novel framework for synergizing MIM and contrastive learning in a close-loop.
2, Improvement over MAE: Better than MAE with shorter training epochs.
Weaknesses
1, Do you need to extract the mask set feature? Will that slow down the training?
2, Why the s means the importance of the patches? Will the global token have global information at the beginning of the training?
3, Can you compare the real training time? Your algorithm seems to be much more complicated than the original MAE. (thus More time over one epoch).
4, Which token do you use for the downstreaming classifcation task?



**Summary Of The Paper:**

1, Adopt importance sampling to select the masked patches with richer semantic information for reconstruction, instead of random sampling as done in previous MIM works.
2, Propose a new contrastive loss that aligns the tokens of the vision transformer extracted from the selected masked patches and the remaining ones.
3, The proposed contextual MIM and contrastive learning are synergetically performed in a loop with fast convergence and strong performance on downstream tasks without ad-hoc augmentations

**Summary Of The Review:**

The method provides a better version of the original MAE with better performance and faster convergence.

---

> ### Author Response · Authors · 2022-11-09
> **Response to Reviewer tKmv**
>
> Thank you for the time and thorough comments, and your concerns inspire us on how to clarify the confusing parts of our work. We clarify your concerns point-by-point as follows:
>
> **Q1 (complexity) and Q3 (computation cost)**. Empirically, since the first selection stage does not require gradient backpropagation, the selection stage can be regarded as inference (usually 3 times faster than training). In practice, ccMIM spends 6 minutes longer than MAE for one epoch. Specifically, ccMIM spends about 22 minutes while MAE spends 16 minutes on 16 V100 GPUs in our machine. However, ccMIM only takes about 6600 minutes (300 epochs * 22) to get the same accuracy as MAE, where MAE spends about 25600 minutes (1600 * 16). Therefore, ccMIM can greatly shorten the time for pretraining.
>
> **Q2 ([CLS] token selection)**. At the beginning of training, the [CLS] token is randomly initialized, so it can’t select contextual-richer patches, and correspondingly, important-sampling-based strategy degenerates into random-based strategy. But after the first few epochs pretraining, the [CLS] token could distinguish which patches include semantic-richer information. For better illustration, we add Figure 3(a) in Appendix in our revised version. Besides, we also illustrate the reconstruction loss curve of MAE and ccMIM in Fig. 3(b) of the first 50 epochs, and the reconstruction loss values of ccMIM are always larger than MAE, which also demonstrates the higher difficulty for reconstruction.
>
> **Q4 (token for downstream tasks)**. For fair comparisons, in line with previous methods (MAE, iBOT, etc.), we use the average pooling of the image patches.

---

> > ### Comment · Reviewer_tKmv · 2022-12-05
> > **Thank you for your response.**
> >
> > Thank you for your clarification.

---

### Official Review · Reviewer_ZDzp · 2022-10-24

**Confidence:** 4
**Correctness:** 3
**Technical Novelty And Significance:** 2
**Empirical Novelty And Significance:** 2
**Recommendation:** 6

**Clarity, Quality, Novelty And Reproducibility:**

- Clarity: The paper is easy to follow despite some details needing clarification (see Weaknesses).
- Quality: The paper addresses a practical problem in an intuitive way, but the experimental results seem insufficient to fully validate the method.
- Novelty: Brings contrastive learning to MIM is already well-studied in many works. The importance-based sampling strategy is new to the self-supervised learning community. But it should be better to discuss such a strategy with contrastive learning.
- Reproducibility: Hyper-parameters and implementation details are all included in the paper.

**Strength And Weaknesses:**

### Strengths

1. The motivation makes sense. The paper writing is easy to understand.
2. Results on image classification / object detection / semantic segmentation all show improvements.

### Weaknesses

1. As shown in Fig.1, the importance score of each patch is computed with the CLS token before feeding into ViT-Encoder. This means the CLS token has no interaction with image patches. So is the importance ranking generated from the CLS token really reliable? More descriptions and visualizations should be added for clear understanding.
2. Bringing contrastive learning to MIM is already well-studied in many works. The more important component in this paper is the importance-based sampling strategy. However, sampling semantic-less patches for harder self-supervised learning objectives is suitable for not only MIM but also contrastive learning methods. It's important to conduct such a sampling strategy to contrastive learning methods (e.g., BYOL [Grill et al., 2020] or SimCLR [Chen et al., 2020b]) and see if consistent performance gains could be achieved.
3. ccMIM feeds both visible patches and masked patches into ViT encoder. The extra computation costs are not mentioned in the paper.
4. Improvements compared to previous methods are a bit marginal.
5. Experiments on larger architectures (e.g., ViT-L) could bring more authority.

**Summary Of The Paper:**

This paper presents a contrasting-aided contextual masked image modeling framework, termed ccMIM. The main motivation behind this paper is to use CLS tokens to mask semantic patches so that the reconstruction target could be more difficult. To make the CLS token contains more semantic context, ccMIM introduces a global alignment loss (InfoNCE or Self-Distillation) on the CLS token. Lots of experiments are conducted to show the superiority of ccMIM.


**Summary Of The Review:**

This paper presents ccMIM. The core component behind ccMIM is to sample semantic-less patches for harder self-supervised learning objectives. Despite better results being achieved, the core contribution of ccMIM is not fully studied (see weakness).

---

> ### Author Response · Authors · 2022-11-09
> **Response to Reviewer ZDzp**
>
> Thank you for your suggestions to help improve the work. Here are our responses to your questions:
>
> **Q1 (Is [CLS] token reliable?)**. [CLS] token is randomly initialized, and if we don’t add the regularization on this token, the importance ranking may not be reliable. However, when we add the contrastive objective (see Eq. 7 and Eq. 8 in our paper) to force the [CLS] token to learn global information on masked patches and visible patches, the [CLS] token could learn global information, which is emphasized in Figure 3(a) in our revised version.
>
> **Q2 (Importance masking for contrastive learning)**. To our best knowledge, except iBOT, no previous contrastive methods use MIM in contrastive loss, and the earlier works use CNN as backbone (e.g., BYOL, SimCLR). Per your request, we use the baseline DINO, and modify it by masking some (0.3 ratio -- see below) important patches whose importance is measured by large attention values with [CLS] token. Specifically, we calculate the contrastive loss on [CLS] token. We train the modified DINO with 100 epochs with mask ratio 0.3 (same as iBOT), and get the following results:
> | Method | linear | finetune |
> | ----- | ----- | ----- |
> | DINO | 67.8 | 79.5 |
> | DINO + Importance mask | 67.9 | 79.7 |
> | MAE | ~ | 79.3 |
> | ccMIM | ~ | **80.5** |
> We can find the mask could serve as an augmentation to further improve the performance, but there are much fewer improvements than those used in MIM method.
>
> **Q3 (Extra computation)**. Compared with MAE, ccMIM feeds visible patches and masked patches independently to the transformer, which brings extra computation. However, ccMIM can spend only 300 epochs to get the performance of MAE with 1600 epochs. We test MAE and ccMIM on 16 V100 GPUs, where MAE takes about 16 minutes per epoch and ccMIM takes about 22 minutes for one epoch. Then, MAE with 1600 epochs could take about 1600 * 16 = 25600 minutes, which is greater longer than ccMIM 300 * 22 = 6600 minutes.
>
> **Q4 (Marginal improvement)**. Actually, compared with other end-to-end methods (only using ImageNet dataset as prior knowledge), our ccMIM gets the highest improvement (0.6 improvements for ccMIM and 0.1~0.2 improvements or less for other methods when finetuning on ImageNet).
>
> **Q5 (Experiments on larger architectures)**. We feel sorry that we can’t provide the experimental results on ViT-L, since it’s quite time-consuming and resource-expensive to conduct experiments on such a huge backbone (MAE uses TPU for pretraining on ViT-L, while this condition is difficult for us). We alternatively select ViT-B with 8*8 patch size as backbone to demonstrate the effectiveness of our method. We hope this could help to address your concerns.
> | Backbone | MAE | ccMIM |
> | ----- | ----- | ------ |
> | ViT-B / 16 | 82.8 | **83.6** |
> | ViT-B / 8 | 82.7 | **84.0** |
> All the results are pretrained with 300 epochs on ImageNet-1K.

---

> > ### Comment · Reviewer_ZDzp · 2022-12-06
> > **The rebuttal addresses my main concerns**
> >
> > Thank you for the new results and the detailed feedback. The rebuttal addresses my main concerns. Despite still a bit of a lack of novelty from my perspective, this work remains a successful practice on the unification of masked image modeling and contrastive learning. I will adjust my rating accordingly.
> >
> > Besides, I believe there are some relevant works that should be cited and discussed in the revision:
> >
> > 1. Kakogeorgiou, Ioannis, et al. "What to Hide from Your Students: Attention-Guided Masked Image Modeling." ECCV 2022.
> > 2. Huang, Zhicheng, et al. "Contrastive masked autoencoders are stronger vision learners." arXiv preprint arXiv:2207.13532 (2022).

---

> > > ### Author Response · Authors · 2022-12-06
> > > **Thanks for your response**
> > >
> > > Thanks for your response.
> > >
> > > The first paper [1] proposes to mask patches with richer contextual information, and integrates the attention MIM in iBOT. Our importance-sampling-based method seems similar to AttMask, where both two works make a small number of semantic-richer patches be visible. However, our ccMIM performs contrasting within the same view, while AttMask requires augmenting twice (well-designed) to generate two views.
> > >
> > > The second paper CMAE [2] is similar to [1]. CMAE also requires to augment twice, and they introduce the two-branch learning framework (i.e., online, target), which we think is a direct way of combining MIM and contrastive learning.
> > >
> > > Both papers [1,2] are relevant to ccMIM, and we would like to cite and discuss the two mentioned papers in our final version.
> > >
> > > [1] Kakogeorgiou, Ioannis, et al. "What to Hide from Your Students: Attention-Guided Masked Image Modeling." ECCV 2022.
> > >
> > > [2] Huang, Zhicheng, et al. "Contrastive masked autoencoders are stronger vision learners." arXiv preprint arXiv:2207.13532 (2022).

---

### Official Review · Reviewer_pWuU · 2022-10-24

**Confidence:** 3
**Correctness:** 4
**Technical Novelty And Significance:** 3
**Empirical Novelty And Significance:** 3
**Recommendation:** 6

**Clarity, Quality, Novelty And Reproducibility:**

- Quality and novelty are sufficient to meet the ICLR standards.
- It is difficult to pinpoint exactly why the method performs well because there are not a few ablation studies needed for the clarity of the methodology.
- As the explanation of  constructing [CLS] token, it is not straightforward to reproduce this method with this manuscript only.

**Strength And Weaknesses:**

* Strength
1. The advantages and disadvantages of the two existing representation learning methods are well described and appropriately utilized in the method design.
2. The mask region separation strategy is novel and appears to be as effective as in tab 5. (The ablation experiments shown in tab 5 are well designed to support the use of the strategy.)
3. The learned representation through the proposed method was effective in all the evaluation tasks which are widely used in verifying the effectiveness of the representation learning.


* Weaknesses
1. Some informative ablation studies may be necessary. E.g., the effect of various masking rate.
2. Detailed information on how the [CLS] token is constructed is lacked or insufficient.
3. It is necessary to experimentally confirm the advantages and disadvantages of the two existing representation learning methods and whether the proposed method has acquired each of these advantages well.

**Summary Of The Paper:**

This paper introduces self-supervised representation learning, which combines two well-known existing methods (i.e., contrast learning and mask autoencoder learning) in a way that appropriately obtains their respective advantages. Contrastive learning with Simese networks is faster in optimization compared to mask auto-encoder training, but is relatively poor at extracting spatially local information. On the other hand, masked encoder learning has performance advantages by making the learned representation have the ability to extract spatially local information, but the optimization is very slow. Since the main structural difference between the two existing methods is the preparation of the input, the proposed method uses visible patches and mask patches as two separate sets of patches required for contrastive learning. This method also proposed a novel mask region separation strategy that makes learning more difficult to obtain stronger representations. This strategy uses importance sampling using weights defined similarly to the attention weights for each value in ViT's self-attention module while input is the additional global token instead of local patches. Experiments have demonstrated that the learned representation through the proposed method (called ccSSL) was effective in terms of classification ability in linear probing task and finetuning task, and transferring ability in multiple downstream tasks.

**Summary Of The Review:**

As this paper introduces a new and effective expression learning method by appropriately utilizing the advantages of the two well-known methods, I am on the positive side and I think it will attract the attention of many researchers.

---

> ### Author Response · Authors · 2022-11-09
> **Response to Reviewer pWuU**
>
> Thank you for the time and thorough reviews. We appreciate your positive comments on our method and experiments. Here are our responses to your questions:
>
> **Q1 (Ablation on masking rate)**. In our initial experiments, we indeed had tested our method with 50%, 65%, 75%, 85% and 95% mask ratios, and we report these results (fine tune accuracy) on ImageNet with 100 epochs:
> | Mask ratio | 50% | 65% | 75% | 85% | 95% |
> | ------ | ----- | ----- | ----- | ----- | ----- |
> | ccMIM | **83.3** | **83.5** | **83.6** | **83.5** | **83.3** |
> | MAE | 82.1 | 82.4 | 82.8 | 82.7 | 82.5 |
> All the results are pretrained on ImageNet with 300 epochs. ccMIM can get more gains at low mask ratios, since ccMIM masks semantic-richer patches, increasing the difficulty of reconstruction for low mask ratios.
>
> **Q2 (How the [CLS] token is constructed)**. The [CLS] token is randomly initialized as most deep learning model does. During the token learning, we add the contrastive loss on [CLS] token to force [CLS] token to learn global information and select important patches (see Eq. 7 and Eq. 8 in our paper).
>
> **Q3 (Comparison with contrastive and MIM)**. As shown in Table 1, our method shows SOTA results by finetuning on ImageNet with 800 epochs, which shows the significant performance advantage of MIM methods. Here, we further experimentally show the convergence advantage of ccMIM, by  listing the finetuning accuracy of 50 and 100 epochs as follows:
> | Epochs | DINO | MAE | ccMIM | Random init |
> | ----- | ----- | ----- | ----- | ----- |
> | 50   | 79.3 | 78.9 | **80.1** | 78.5 |
> |100   | 79.5 | 79.3 | **80.5** | 78.5|
> where our method gets the highest results compared with the contrastive method (DINO) with the first 50 and 100 epochs pretraining.

---

### Author Response · Authors · 2022-11-09
**General Response**

**Dear Area Chair and Reviewers:**

We would like to express our sincere thanks to the reviewers' for their time and valuable suggestions. Overall, we are encouraged that most reviewers acknowledged the paper’s novelty (pWuU, ZDzp, tKmv), Quality (pWuU, tKmv) and positive experimental results (pWuU, ZDzp, tKmv).

---

### Author Response · Authors · 2022-11-18
**Look forward to reply to our response and update**

Thanks to the efforts of ACs and all reviewers. Approaching the pdf updating DDL, is there anything needing added?

---

### Decision · Program_Chairs · 2023-01-20

**Decision:**

Accept: poster

**Justification For Why Not Higher Score:**

The methodology is somewhat incremental, since it combines two previous works, MIM and contrastive learning.

**Justification For Why Not Lower Score:**

Although the novelty is somewhat incremental, combining contrastive learning and MIM is done in a clever way, to achieve synergy between the two methods, and the results are good.  MIM and Contrastive learning are two important branches of self-supervised learning, and their combination is interesting.

**Metareview: Summary, Strengths And Weaknesses:**

**Summary:** This paper introduces self-supervised representation learning, which combines two well-known existing methods (i.e., contrast learning and mask autoencoder learning) in a way that appropriately obtains their respective advantages. The main motivation behind this paper is to use CLS tokens to mask semantic patches so that the reconstruction target could be more difficult. To make the CLS token contains more semantic context, ccMIM introduces a global alignment loss (InfoNCE or Self-Distillation) on the CLS token.

**Strengths:** The motivation makes sense. The paper writing is easy to understand. Results on image classification / object detection / semantic segmentation all show improvements.

**Weaknesses:** The following major concerns were identified by the reviewers:

1. missing some ablation studies (e.g., effect of mask rate) [pWuU]
2. missing information about how [CLS] token is constructed [pWuU]
3. need to confirm in experiments whether the advantages/disadvantages of the 2 representation learning methods is present in the proposed combination model. [pWuU]
4. CLS token has no interaction with image patches. Is it reliable? [ZDzp]
5. the core part is the sampling strategy. but there are missing experiments using the same patch sampling strategy for other CL methods, to show the effectiveness of the sampling strategy. [ZDzp]
6. what is the extra computation cost? [ZDzp, tKmv]
7. marginal improvement compared to previous methods. [ZDzp]
8. missing experiments on larger architectures (ViT-l) [ZDzp]
9. Why is s the importance of the patch? [tKmv]
10. Will the global token have global information at beginning of training? [tKmv]
11. Which token is used for downstream classification? [tKmv]



**Note From Pc:**

if the above contains the word "oral" or "spotlight" please see: "oral" presentation means -> notable-top-5% and "spotlight" means -> notable-top-25%. As stated in our emails, we are disassociating presentation type from AC recommendations

**Summary Of Ac-Reviewer Meeting:**

During the discussion, 2 out of 3 reviewers were satisfied with the response, while 1 reviewer did not participate. The AC checked and the missing reviewer's comments were addressed well in the response. In the end, Reviewer ZDzp upgraded their rating, with the paper finally receiving all 6 ratings.